# Polishing of metal 3D printed parts with complex geometry: Visualizing the influence on geometrical features using centrifugal disk finishing

**Kirsten Lussenburg****[1]\*, Remi van Starkenburg[2], Mathijs Bruins[1], Aimée Sakes[1], Paul Breedveld[1]**

**1** Bio-Inspired Technology Group (BITE), Department BioMechanical Engineering, Faculty of Mechanical, Maritime, and Materials Engineering, Delft University of Technology, Delft, The Netherlands, **2** Department of Electronic and Mechanical Support Division (DEMO), Delft University of Technology, Delft, The Netherlands

\* k.m.lussenburg@tudelft.nl

## Abstract

Parts produced with metal additive manufacturing often suffer from a poor surface finish. Surface finishing techniques are effective to improve the quality of 3D printed surfaces, however they have as downsides that they also slightly change the geometry of the part, in an unpredictable way. This effect on the geometrical features of complex parts has received little attention. In this research, we illustrate a method to visualize the impact of surface finishing techniques on geometrical features, as well as their effectiveness on parts with high shape-complexity, by using centrifugal disk finishing as a case study. We designed and 3D printed test parts with different features using selective laser melting, which were coated with a blue metal lacquer prior to polishing. After polishing, the blue lacquer was eroded away from the spots that were easily reached by the polishing process, yet had remained on the surfaces that could not be reached by the process. We used measurements of material removal and image processing of the remaining blue lacquer on the surfaces to analyze these effects. Using this method, we were able to derive a number of specific design guidelines that can be incorporated while designing metal AM parts for centrifugal disk finishing. We suggest that this visualization method can be applied to different polishing methods to gain insight into their influence, as well as being used as an aid in the design process.

## 1. Introduction

Metal additive manufacturing (AM) processes allow for the production of components with a high shape complexity and with excellent mechanical properties. Most current metal AM processes use a metal powder that is fused together using a laser in the desired shape, which is referred to as powder bed fusion (PBF). Widespread adoption of metal AM for precision applications is hindered by the rough surface finish of as-built parts [1], caused by the powder-based nature of the process [2]. For high-precision mechanical parts, such as those used in

**Data Availability Statement:** All image files and Matlab script are published and can be accessed through the following DOI: 10.4121/f124ec67-7c9c-4814-9b18-2a67e4a673b1.v1

**Funding:** This project has received funding from the Interreg 2 Seas programme 2014-2020 co-funded by the European Regional Development Fund under subsidy contract No. 2S04-014: https://www.interreg2seas.eu/nl/3dmed. The funders had no role in study design, data collection and analysis, decision to publish, or preparation of the manuscript.

**Competing interests:** The authors have declared that no competing interests exist.

medical instrumentation, a poor surface quality increases friction and wear on parts that interact with each other [3, 4]. Relatively speaking, the influence of a poor surface quality is even more pronounced for small or miniaturized parts, where geometrical feature sizes are only a few times larger than the size of the metal particles. An example of this is shown in Fig 1.

Surface finishing techniques can be employed to improve the surface quality of these parts, by removing partially adhered particles from the surface, closing pores and microcracks, and diminishing visible layers and laser patterns [5–7]. Mechanical surface finishing techniques, in which a mechanical interaction with the surface is responsible for the polishing rather than a chemical or electrical interaction, are reported to be most effective for creating a high quality surface [8–10]. With mechanical polishing a mirror-like surface can be produced (0.0254 μm) [11]. However, when applied to complex AM geometries, most mechanical processes have the disadvantage that they are extremely labor-intensive and require a lot of manual intervention [8], or they may not be able to reach all surfaces of the object at all. Therefore, they may be less suitable to process functional, complex 3D printed parts in a high volume [12], and they negate the advantages of the high shape complexity enabled by AM.

Mass Finishing (MF) processes offer a solution for these applications [12, 13]. The working principle of MF is based on the use of abrasive polishing media, such as ceramic or plastic beads, agitated by mechanical means, often by rotation or vibration. The polishing media, in some cases combined with a liquid medium, comes into contact with the surface of the submerged parts and removes material from the surface by abrasion, as well as impart a polishing effect. Important processing parameters for MF processes are the size of the polishing media, speed of motion, and processing time. Generally, the use of larger sized particles results in a higher material removal rate and a better surface quality [14, 15]. Prolonged contact with the abrasives in the form of a longer processing time results in the same effect [14, 16, 17], up to a limit [15]. Higher speeds are also associated with higher material removal rates, due to higher forces being applied to the workpieces [16, 18, 19]. The fill level of the container with polishing media was shown to have a negligible effect on material removal rates, higher load levels decreased the chance of contact between the abrasives and workpiece [14, 18]. The advantages of MF processes are that the process is hands-off, it can process multiple parts at once, and it can be customized to the type of metal and surface finish required.

When the settings of MF processes are properly selected, surfaces with Ra of 0.52 to 5 μm Ra can be obtained [14]. However, as with any surface finishing process, unwanted side effects of MF processes include rounding of sharp edges, changes in part dimensions, decreased feature resolution, and changes in flatness [11, 20]. These side effects are especially impactful for miniature parts, and become more prominent with longer processing times [13]. Surface improvements for internal surfaces are also less pronounced, unless they are designed significantly larger than the polishing media [18].

So far, research efforts into surface finishing processes for metal AM parts have focused on illustrating the improvements in surface finish only, executed on simple geometrical shapes such as cubes or discs [4, 9, 10, 12, 21–24]. As such, little information on the geometrical impact of surface finishing techniques on complex structures, such as those that can be produced with AM, is available. To apply PBF-printed parts in miniature, high-precision applications, it is important to not only know the surface roughness improvements that can be obtained with various methods, but also how effective these methods are on complex features, as well as the geometrical effects on features. This information can aid in the development of more robust design guidelines, facilitating higher quality in the AM process and the final components [13].

Making informed decisions early on in the design process is already commonly applied in design methodologies such as Design for Additive Manufacturing [25], although no such

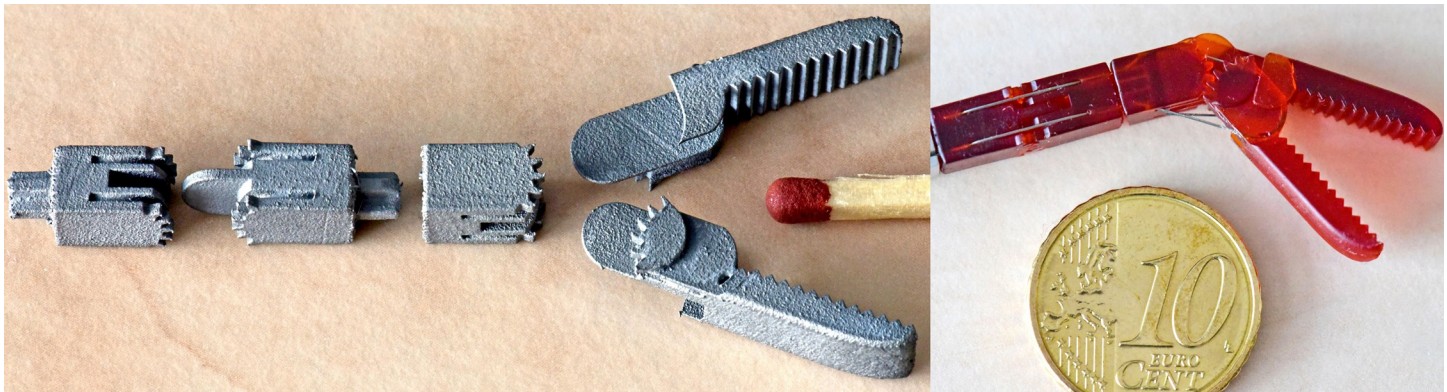

**Fig 1.** Example showing the rough surface of parts printed with metal additive manufacturing (left). The parts are meant to be used in a steerable, medical instrument, however they require extensive polishing before they can be used in a functional mechanism. The match is shown for scale. As comparison, on the right the same parts 3D printed using a polymer-based additive manufacturing process.

guidelines currently exist for post-processing methods. Therefore, in this study, we investigate how geometrical features are altered as a result of centrifugal disk finishing (CDF), a common MF process, and which types of features do not lend themselves well to polishing by CDF. In order to do so, we designed an experiment in which a miniature 3D printed test part witch complex geometrical features was coated with a blue marking lacquer for metals, before undergoing polishing. After polishing, the remaining lacquer gives a visual pattern of the surfaces on the part to which the polishing media had access. The results led to a number of design guidelines that can be of help during the design process of SLM parts.

## 2. Materials and methods

### 2.1 Process

For this study, we designed a test part as shown in Fig 2A, which includes extruding features, recessed features, and mechanical features (gear teeth, ledges). The features were designed in such a way that none of them were obscured by other features, and all were large enough to allow theoretical access of polishing media. Twelve test parts were 3D printed by a commercial company, Materialise NV (Leuven, Belgium) on an EOS M280 (EOS GmbH, Germany), in stainless steel 316L. The parts were printed in the upright position shown in Fig 2A. Parts received a heat treatment to reduce internal stresses and were removed from the support structures using wire electrical discharge machining. The specific print and processing settings used for the test parts were not disclosed by the manufacturer. Instead, as a reference value, the surface roughness of the flat side on one of the test parts was measured (Mitutoyo SJ-301 Surftest, Mitutoyo Corporation, Japan) with a tolerance of 0.01 μm, across a length of 4 mm. An example of a printed test part is shown in Fig 2B.

Surface roughness is often used as a measure of the effectiveness of surface finishing processes, however this is difficult to measure on complex geometrical shapes. Since not all surfaces will be targeted equally, this would also require many measurements. In contrast, with the naked eye it is difficult to see which surfaces have been targeted by the polishing process and to what amount. The solution we found is to apply a blue marking lacquer for metals (Griffon, Bolton Adhesives, The Netherlands) on the parts that is eroded away in places that make contact with the polishing media. The coating will remain in places that are not in contact, or not in sufficient contact, with the media, leaving a visual pattern on the surface. The lacquer was applied on the parts after printing, with an approximate thickness of 0.05 mm. It

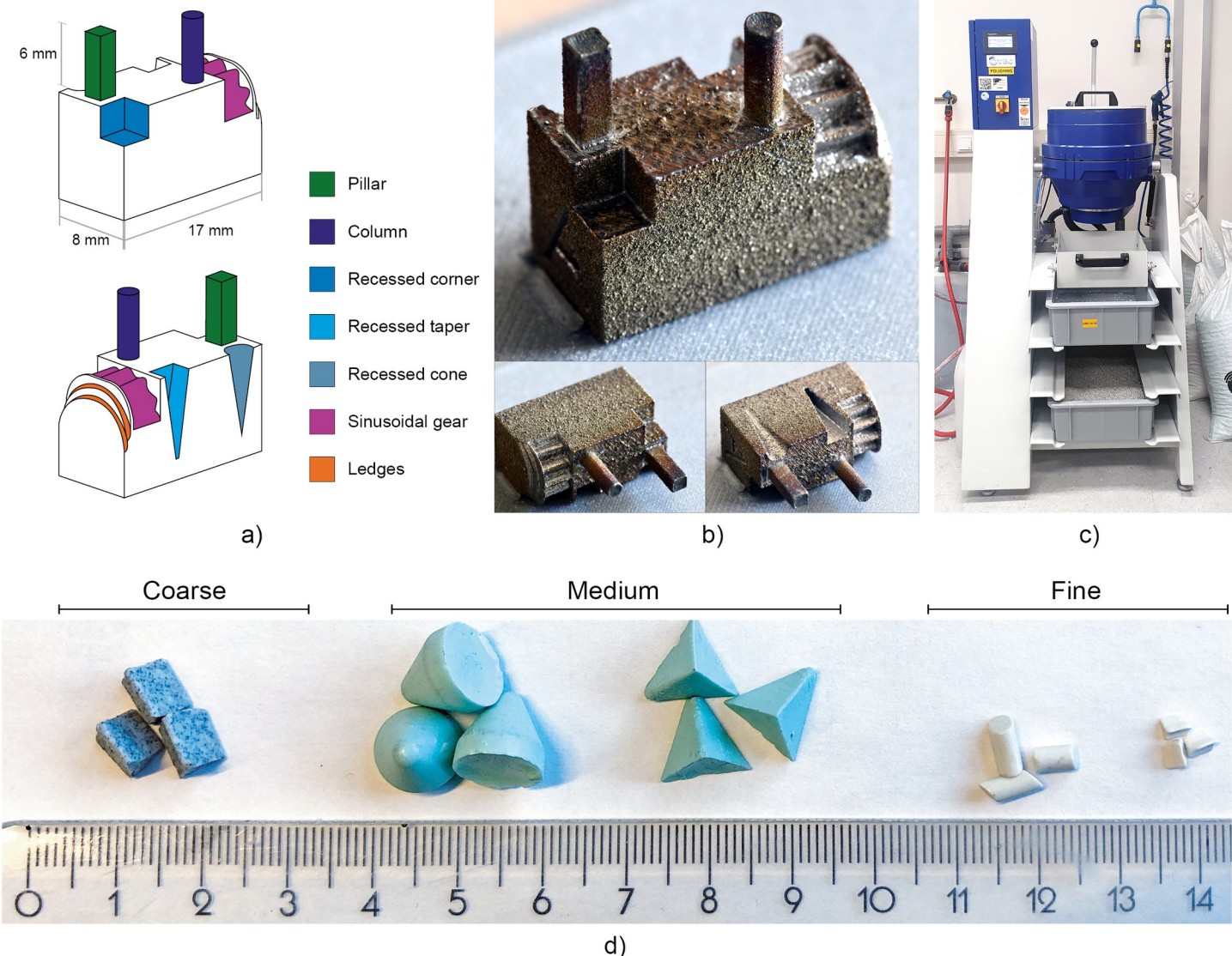

**Fig 2. Test parts used in this study.** a) The designed test part showing the different features that were implemented. b) An example of one of the test parts after printing. c) The CDF used in this study. d) The different types of polishing media used in this study. Coarse: DZS 6/6, medium: KM 10 and PM 10, fine: ZSP 3/5 and DZP 3/3 SK (OTEC, Germany).

should be noted that since the lacquer influences the effectiveness of CDF, a smaller improvement in surface finish will be expected.

The parts were polished using CDF (CF 1x18 B, OTEC, Germany), shown in Fig 2C. In CDF, the parts are submerged in a container with polishing media with or without a liquid, which is brought into motion by a rotating disk at the bottom. Several steps with different polishing media are usually applied for the best results. Generally speaking, larger abrasive media will be more effective on the external surfaces of the part and have a faster cut rate, while smaller media are able to reach into the interior regions and small features [11, 26]. The size of the abrasive media is also important to keep the submerged parts separate from each other and prevent them from clashing [13]. The shape of the abrasive media should be chosen in such a way that it permits access to all surfaces of the part [13]. Based on preliminary tests and advise

of the manufacturer of the CDF, we settled on a polishing schedule consisting of three steps of 120 minutes with coarse, medium, and fine polishing media with different shapes. In Table 1, the details of the used process steps are given, for each step fresh media was used. We investigated the influence of the coarseness of the media and the total duration of polishing on the geometrical features, by dividing the twelve parts into four groups of three that were polished in different polishing steps, as shown in Table 2. The order of the steps was in decreasing coarseness in all cases. The polishing media is shown in (Fig 2D). It should be noted that while the KM 10 and PM 10 media are recommended as a second step for a less abrasive polish, the particles itself are larger in size than the coarse media.

## 2.2 Analysis

As a measure of the overall effectiveness of the CDF process, we measured the removed material on different places on the part. No additional surface roughness measurements were performed, since the applied lacquer would interfere with the measured roughness values. The overall width of the part, as well as the top and the base of the pillar and column, were measured before polishing and after each polishing step using a digital micrometer (Mitutoyo Digimatic IP65 0-25mm, Mitutoyo Corporation, Japan), across the entire length of the parts. The radius of one of the outer edges was measured using a digital microscope (Dino-Lite 3.0, AnMo Electronics Corporation, Taiwan). For the recessed taper and cone, using the digital microscope we determined how deep into the feature the polishing media had reached, by measuring the distance from the top of the feature to the edge of where the blue lacquer was completely intact.

Images of different surfaces and features were used to visually compare where the coating was eroded away and with what intensity. To analyse the remaining blue colour on the parts, photos were taken of the sides of the test part and of the features using a digital microscope in standard office lighting (Fig 3A). The images were processed in Photoshop (Adobe Inc., USA) to extract only the blue tones. An RGB image contains three channels: red, green, and blue. The 'Color Range' command allows the user to select pixels of an image based on their colour channel. By selecting 'Blues' within the Color Range command, only pixels within the blue channel were selected. All other pixels were set to a white colour (Fig 3B). These images were saved as Bitmap images and processed in MATLAB R2021b (The MathWorks Inc., USA), using a script that converts the blue pixels to black, using the function *im2bw* with a threshold of 0.8 (Fig 3C) [27, 28], after which the percentage of black pixels of the total pixels was calculated for each image. The areas that were analysed were one of the side surfaces, including the recessed corner, the top surface between the extruding features, the top view of the gears and ledges, and one side of the extruding pillar and column (Fig 3D).

# 3. Results and discussion

## 3.1 Surface effects

The initial surface roughness on the side of the test part was Ra = 9.49 μm before polishing, which is in line with reported roughness values between 2 and 15 μm Ra for SLM parts [14, 29, 30]. Fig 3A shows the normalized measurements of the width of the test parts before and after polishing, compared with the as-drawn width of the test part (horizontal axis). The average width of the as-printed test parts was 8.17 mm ± 0.05 mm before polishing, compared to a designed width of 8.0 mm. After polishing, the average width of the parts in Group 1 to 4 was: 8.02 ± 0.02 mm, 7.94 ± 0.02 mm, 7.97 ± 0.01 mm, and 7.97 ± 0.01 mm, respectively. These results show that the CDF process can be used to correct for the dimensional error caused by the printing process. It should be noted that these measurements include the thickness of the

**Table 1. Process details for the different centrifugal disk finishing polishing steps used.** All materials are from OTEC, Germany.

|  | Coarse step | Medium step | Fine step |
|---|---|---|---|
| Polishing media | DZS 6/6 | KM 10 and PM 10 | ZSP 3/5 and DZP 3/3 SK |
| Size | 6 x 6 mm | 10 x 12 mm and 10 x 10 mm | 3 x 5 mm and 3 x 3 mm |
| Compound | SC15 | SC15 | SC5 |
| Speed | 280 rpm | 260 rpm | 220 rpm |
| Water flow | 10 L/h | 10 L/h | 10 L/h |
| Water concentration | 3% | 3% | 3% |
| Type | Ceramic-bonded | Plastic-bonded | Ceramic-bonded |

layer of lacquer of approximately 0.05 mm that was partially polished away. Fig 4B shows that for Group 1 and Group 2 on average 0.21 mm material was removed from the width, for Group 3 on average 0.23 mm, and for Group 4 on average 0.14 mm. It is clear that the coarse polishing step is responsible for most of the material removal, which is in agreement with other studies [14], although for Group 4 it is noticeable that the medium step has more influence on the total material removal than for Groups 2 and 3. This implies that the first polishing step is responsible for most of the material removal, regardless of the coarseness of the polishing media. Rounded edges are a familiar side effect of mechanical polishing that is also present in our test parts. Only edges on the outside of the part (positive edges) experience this effect, because theses edges are more exposed to the polishing media. All sharp outside edges have been rounded by the polishing process, while the internal edges were not polished. The average outer edge radius was 0.23 ± 0.03 mm after polishing. Group 4 experienced the smallest edge rounding with a radius of 0.20 ± 0.03 mm.

In all instances, after polishing blue lacquer has remained in the 'pores' on the surfaces of the parts (Fig 5). The calculated percentages of blue color on the analyzed top and side areas are given in Table 3 and Fig 4C–4D. For the side of the part, the amount of remaining lacquer decreases from Group 1 to Group 3, as expected with additional polishing steps. The most lacquer remained on the test parts in Group 4, illustrating that one course polishing step removes more material than a medium and fine step together. When comparing Fig 4B–4D, it can be seen that the remaining lacquer on the surface decreases with additional polishing steps, i.e. the surface quality increases, however for the material removal this connection is less distinct. This shows that the influence of additional, finer polishing steps on the amount of removed material stagnates, while the surface quality keeps improving. This is in line with other research suggesting larger particles are responsible for most of the material removal [14, 15].

On the top surfaces, it is visible that the extruding features and protrusions have partly shielded the surrounding areas from being polished (Fig 5A). This is also clearly visible in the calculated blue color in the area between the extruding features in Fig 4D. The average of the remaining color for each group is higher than that for the side surfaces. This is most noticeable for Group 4, where the average percentage of remaining color is more than twice as high as on the side surface. A possible explanation is the size of the medium media, which is larger than

**Table 2. The polishing steps and time for each of the test groups.**

|  | No. of parts | Coarse | Medium | Fine | Total time |
|---|---|---|---|---|---|
| Group 1 | 3 parts | 120 min | - | - | 120 min |
| Group 2 | 3 parts | 120 min | 120 min | - | 240 min |
| Group 3 | 3 parts | 120 min | 120 min | 120 min | 360 min |
| Group 4 | 3 parts | - | 120 min | 120 min | 240 min |

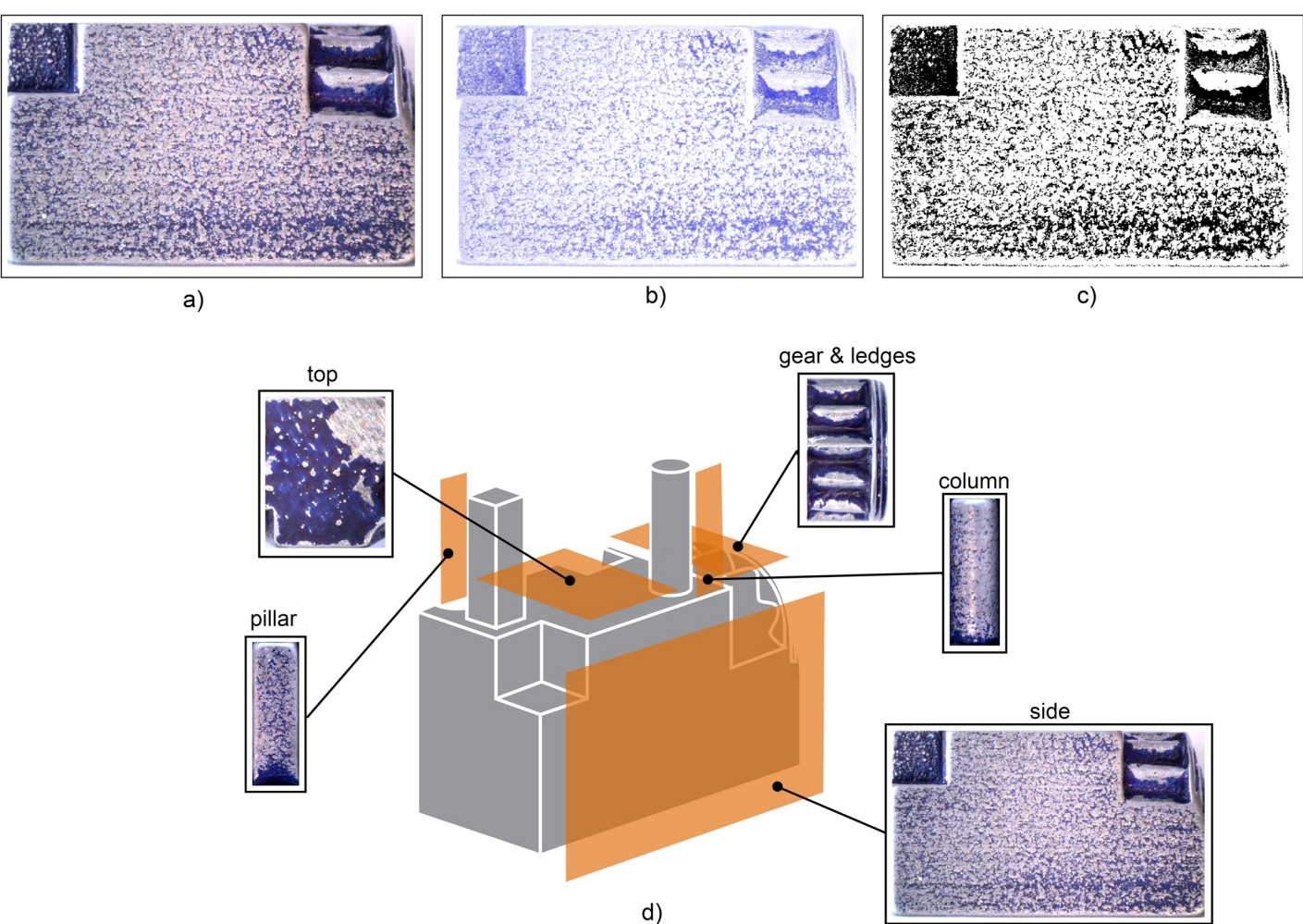

**Fig 3. Image processing of the remaining blue color on different surfaces on the parts.** a) Photo taken of the side of one of the test parts, shown here as an example. b) The same image with only the blue tones extracted. c) The image converted to black and white, in which the blue pixels are converted to black. These image were used to calculate the percentage of remaining blue on the surfaces. d) Location of the areas that were analyzed indicated in orange, with example photos of one of the test parts.

the coarse media and therefore does not fit as well between the features. However, it is noticeable that this effect does not occur for Group 2, which has a significantly lower percentage blue left than Group 1. It is possible that the coarse media partly dislodges the lacquer, which is subsequently removed by the medium and fine media, while on their own the medium and fine media are not abrasive enough to remove the lacquer between the extruding features.

## 3.2 Feature effects

In Fig 5A and 5B, it is visible that more lacquer remains on the recessed features than on the extruding features. The mechanical features, i.e. the gear teeth and ledges, show a lot of remaining lacquer, indicating that the polishing media does not properly access them, although in theory the features were large enough to allow access. Fig 6A shows the calculated percentage of blue in the gears and ledges combined. Here the remaining lacquer is higher than for the flat surfaces, and there are no substantial differences between the four groups. Adding more polishing steps seems to have no significant advantage for the polishing of these

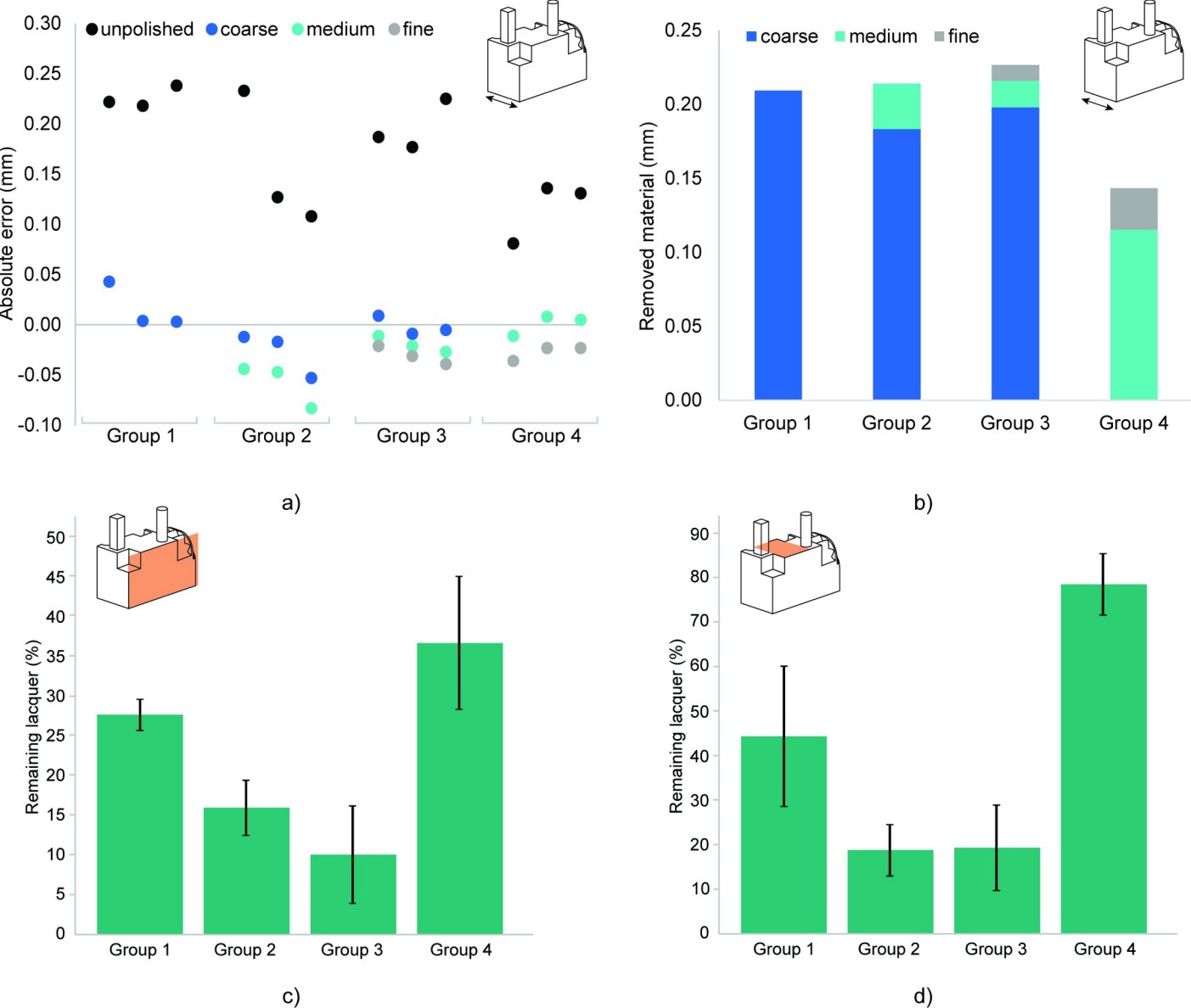

**Fig 4. Material removal caused by polishing.** a) Absolute dimensional error of the width of the test parts. b) Material removal of the width of the parts per polishing step. c) Calculated percentage of remaining blue lacquer on the side of the part. d) Calculated percentage of remaining blue lacquer on top of the part between the two extruding features.

features, which remain hard to access. Similar results were found for the recessed cone and taper. The extent to which these were polished was measured by the depth of the feature to which the polishing media reached. For Groups 1 to 3, the cone was considerably more polished than the taper, with a depth of 3.67 ± 0.21 mm versus 3.04 ± 0.04 mm for Group 1, 4.39 ± 0.51 mm versus 2.69 ± 0.40 mm for Group 2, and 3.96 ± 1.15 mm versus 3.01 ±0.16 mm for Group 3. In Group 4, none of the recessed features were polished beyond the top edge of the feature. In Fig 5, it can also be seen that more lacquer remains on the sides of the recessed features.

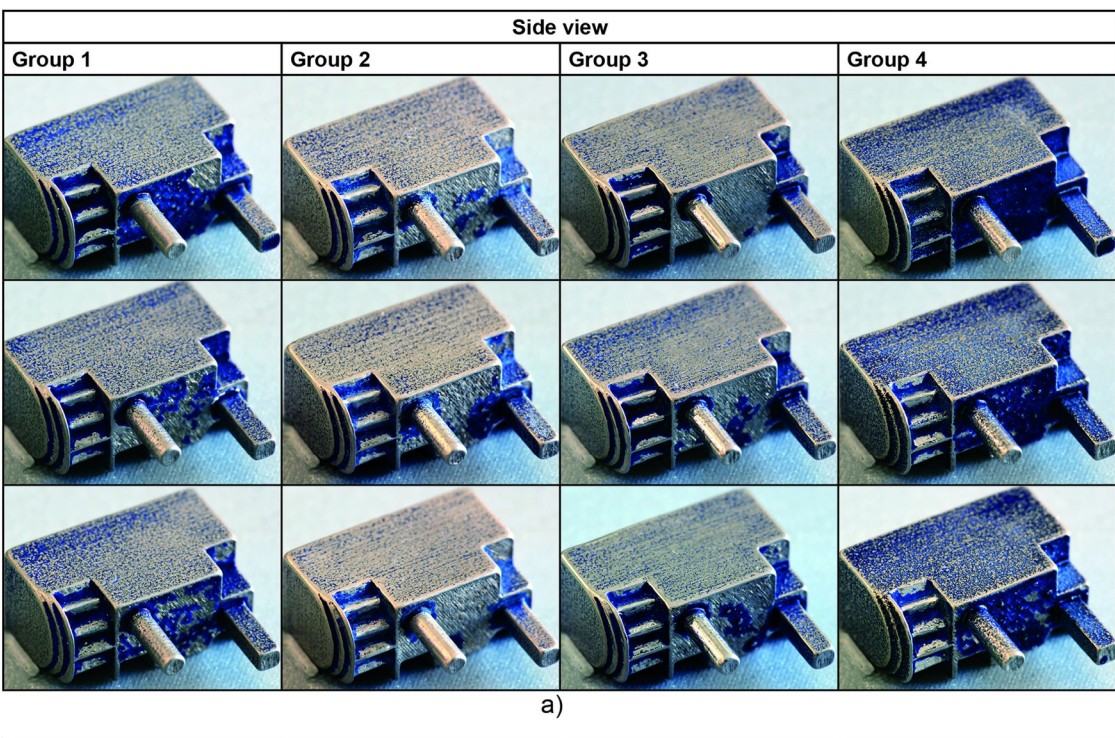

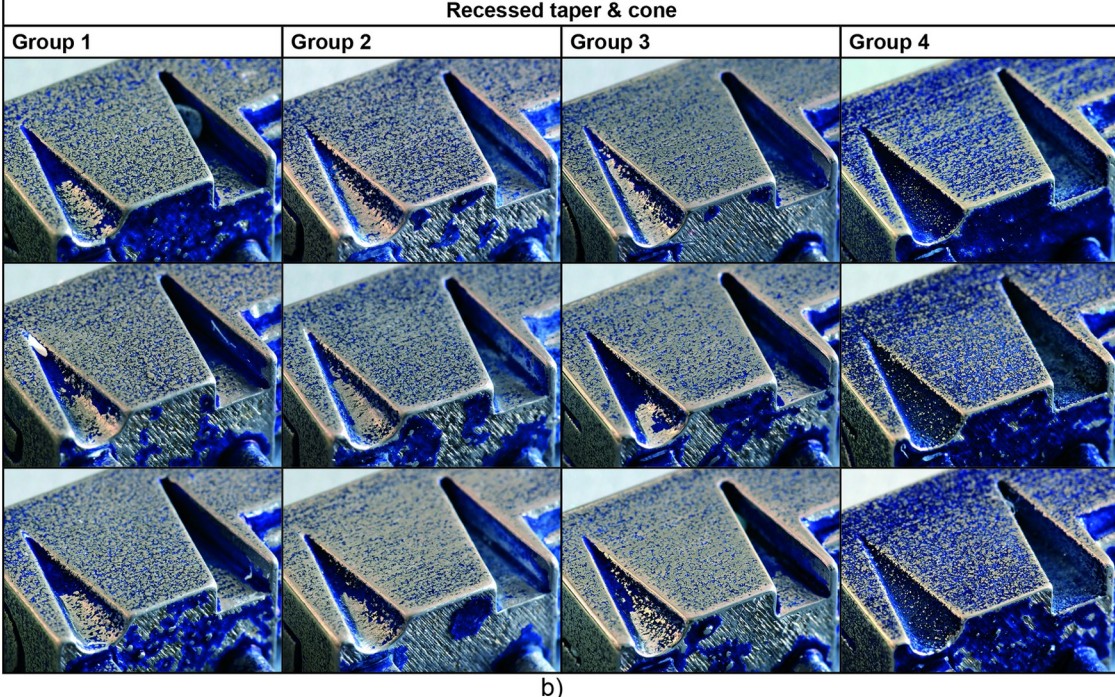

**Fig 5. Remaining blue lacquer on the test parts after polishing.** a) Side view of all test parts per group. b) Close up of the recessed taper and cone on the other side of the test parts.

Fig 5A shows that the protruding columns appear more polished than the protruding pillars, which is confirmed by the calculated percentage of blue in the images (Fig 6B). However, the average material removal, measured at the cross-section of the top of both features is

**Table 3. Calculated percentage of blue lacquer remaining on the parts, based on different views of the test part.**

|  | Group 1 | Group 2 | Group 3 | Group 4 |
|---|---|---|---|---|
| Side | 27.6 ± 2.0 | 15.9 ± 3.5 | 10.1 ± 6.1 | 36.6 ± 8.4 |
| Top | 44.3 ± 15.8 | 18.7 ± 5.9 | 19.2 ± 9.7 | 78.4 ± 7.0 |
| Gear & ledges | 59.2 ± 0.8 | 63.5 ± 6.8 | 56.2 ± 5.2 | 62.6 ± 6.7 |
| Pillars | 26.7 ± 2.0 | 25.4 ± 3.4 | 19.6 ± 6.4 | 38.4 ± 10.9 |
| Columns | 19.1 ±4.0 | 17.3 ± 6.3 | 10.1 ± 2.7 | 22.1 ± 3.1 |

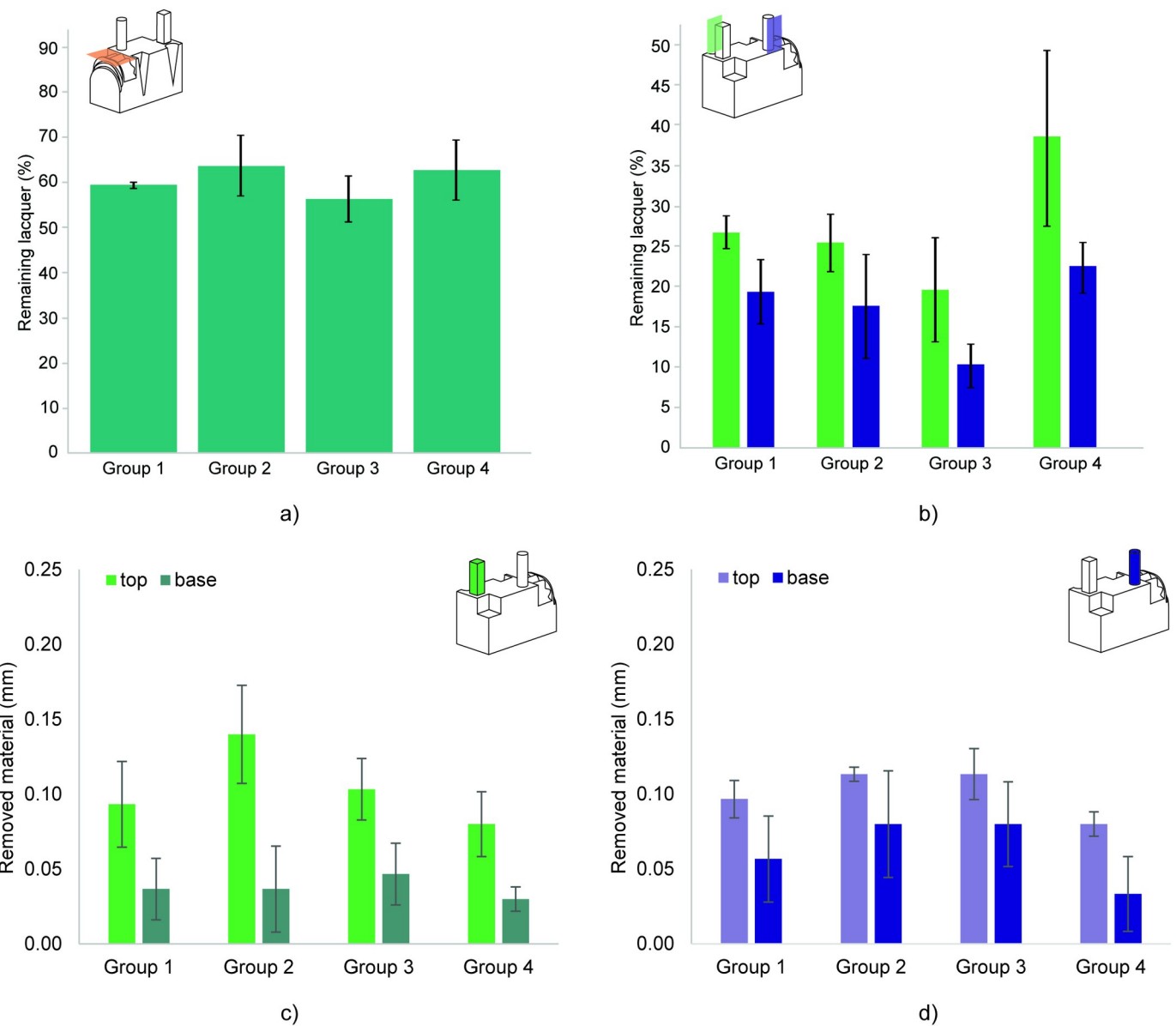

**Fig 6. The effects of polishing on different features of the test part.** a) Calculated percentage of remaining blue lacquer on the gear and ledges. b) Calculated percentage of remaining blue lacquer on the extruding pillar (green) and column (purple). c) Material removal of the pillar, on the top and the base. d) Material removal of the column, on the top and base.

comparable: 0.10 mm ± 0.02 for the column, and 0.10 mm ± 0.03 for the pillar. This indicates that rounded features lend themselves better to polishing than similar angular features. When comparing the material removal at the top with the material removal at the base of both features, as shown in Fig 6C and 6D, for both features material removal at the top was on average 50% more than at the base. Protruding surfaces have a higher chance of being in contact with the polishing media, which means that the diameter of such features will change along its length.

### 3.3 Limitations and recommendations

Visualization of polishing patterns using marking lacquer is a novel approach to further understanding the details of MFT processes. It is a simple and inexpensive method to observe differences in polishing compatibility for different geometries. However, there are still some limitations to this test. For instance, the marking lacquer was difficult to apply evenly across the part, and tended to pool in inside corners and the base of the gear teeth, leaving a thicker layer which could have influenced results. Different marking lacquers or coatings can be tested to obtain a more even application. In addition, the presence of the lacquer on the part will have influenced the effectiveness of the polishing process to some extent. Although the lacquer is less hard than the stainless steel of which the parts are made, this is still an additional layer that needs to be polished away. Therefore, it is an effective measure of visualizing the polishing patterns, but less suitable to determine the actual efficiency of the polishing process with regards to the surface roughness and the material removal rate. The image analysis method employed in this study is useful for comparison within batches, but due to sensitivity to lighting the absolute results may vary in different conditions. Additional research into other process parameters of CDF, such as rotational speed, and abrasive media shape and size, is required to provide insight into their effect on surface finish as well as geometrical features. We recommend that in order to optimize the processing settings for a specific design, test parts with representative features of the final design should be used [26].

Although in this study we focused on the CDF process, we imagine that the same workflow can be applied to different polishing methods in order to gain insight into the effect of the polishing method on the geometrical features and its effectiveness on a complex-shaped part. This method can be particularly helpful to design small, functional parts. Before committing to a final design, we imagine a test part with similar features should be designed and printed, and different polishing steps can be applied. This method can gain insight into which features are problematic for polishing, and should therefore be altered or avoided in the final design, as well as gain insight in which combination of polishing steps and duration is most suitable for the final part.

## 4. Conclusion

From the performed experiments, we can distill some general design guidelines that can be useful when designing metal AM parts that should be processed using CDF.

1. Extruding features shield the remaining surfaces from polishing, and should therefore not be placed next to functional surfaces or next to other extruding features.

2. The discrepancy between top and base of extruding features can be solved in the design phase by applying a negative draft angle to the feature.

3. Recessed features are difficult to polish, although the backside of these features is more easily accessible than the sides. When a functional surface is required in a recessed feature, the back surface of this feature is more suitable than the sides.

4. Rounded features lend themselves better to polishing than features with sharp corners, whether they are extruding or recessed.

5. All extruding edges will be rounded by the CDF process, while recessed edges remain largely unpolished. This should be taken into account when relying on sharp edges for functional features. Since the exact amount of material removed from the edge cannot be controlled during polishing, it is better to apply a rounded edge in the design phase with a known value. Negative edges can be included in the design, although it should be noted that they will receive little polishing.

6. Even if complex features, such as the gears in this study, are designed to allow access to the polishing media, they remain difficult to polish no matter the polishing cycles of CDF. Therefore, these should either be avoided or polished using a different polishing process.

7. Additional polishing steps with fine media appear to have more influence on the surface quality than on the material removal, for the surfaces that can most easily be reached by the polishing media. Therefore, when a mirror-smooth surface is required, it should be sufficient to only take into account the material removal of the first polishing step in the design phase.

## Author Contributions

**Conceptualization:** Kirsten Lussenburg, Mathijs Bruins, Paul Breedveld.

**Data curation:** Kirsten Lussenburg, Remi van Starkenburg, Mathijs Bruins.

**Formal analysis:** Kirsten Lussenburg, Remi van Starkenburg, Mathijs Bruins.

**Funding acquisition:** Paul Breedveld.

**Investigation:** Kirsten Lussenburg, Remi van Starkenburg.

**Methodology:** Kirsten Lussenburg, Remi van Starkenburg, Mathijs Bruins.

**Supervision:** Aimée Sakes, Paul Breedveld.

**Validation:** Kirsten Lussenburg.

**Writing – original draft:** Kirsten Lussenburg.

**Writing – review & editing:** Remi van Starkenburg, Mathijs Bruins, Aimée Sakes, Paul Breedveld.

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
