## [Decision Letter · Decision Letter 0]

7 Dec 2022

PONE-D-22-29565Polishing of metal 3D printed parts with complex geometry: visualizing the influence on geometrical features using centrifugal disk finishingPLOS ONE

Dear Dr. Lussenburg,

Thank you for submitting your manuscript to PLOS ONE. After careful consideration, we feel that it has merit but does not fully meet PLOS ONE’s publication criteria as it currently stands. Therefore, we invite you to submit a revised version of the manuscript that addresses the points raised during the review process.

We look forward to receiving your revised manuscript.

Kind regards,

Amitava Mukherjee, ME, Ph.D.

Academic Editor

PLOS ONE

Journal Requirements:

Reviewers' comments:

Reviewer's Responses to Questions

**Comments to the Author**

1. Is the manuscript technically sound, and do the data support the conclusions?

Reviewer #1: Partly

Reviewer #2: Yes

2. Has the statistical analysis been performed appropriately and rigorously? 

Reviewer #1: No

Reviewer #2: Yes

3. Have the authors made all data underlying the findings in their manuscript fully available?

Reviewer #1: No

Reviewer #2: Yes

4. Is the manuscript presented in an intelligible fashion and written in standard English?

Reviewer #1: Yes

Reviewer #2: Yes

5. Review Comments to the Author

Reviewer #1: The paper focuses on the polishing of SLMed parts by using CDF apparatus. The work shows many lacks in the description of the methodology and the method itself. In following the list of them.

Introduction: some works on the mass finishing applied to SLM parts are neglected. Interesting papers provide basic understanding of the mechanisms and the attainable roughness.

Material and Methods: since the previous observation, the line 103 should be revised adding information coming from the references.

Material and Methods: the use of lacquer leaves many doubts about the media to surface interactions at the beginning of the operation. The CDF is a delicate operation with relatively low energy and activation mechanisms (chipping, plastic deformations, microcracking, microfatigue) may be plagued, modified or postponed.

Material and Methods: authors didn’t provide an experimental plan to find polishing parameters such as: media shape size and material, compound amount and type, rotational speed, water flow, filling percentage, etc. The manufacturer information and experience are generally related to 316L parts fabricated via traditional technologies and different results are expected on the SLMed material completely different in microstructure.

Results and discussion: The previous observation can explain the Ra results of this work. The obtained values after the polishing reveal that the selected processing parameters are not really efficient with respect to other mass finishing operations applied to the same SLM material.

Results and discussion: The assessment at line 190 should be verified.

Material and Methods: The experimental campaign is poor and extraction at intermediate times is general requested for a deeper investigation of the surface evolution. The image processing technique lacks in repeatability with the following issues. The using of a Photoshop algorithm does not allow a clear understanding of the RGB channels selection criteria. Moreover, when a color selection is adopted, particular care must be paid on the light source which may modify the radiation reflection spectrum and have local and surrounding undesired effects on the under-detection feature. Finally, no indication or motivation is provided about the selection of the threshold in Matlab.

Reviewer #2: Comments

In this paper authors used the centrifugal disk finishing process for finishing of o improve the quality of 3D printed surfaces. The authors designed and 3D printed test parts with different features using selective laser melting, which were coated with a blue metal lacquer prior to polishing. The surface improvement is observed through Mitutoyo SJ-301 Surftest instrument.

1. The authors have not added the image of CDF instrument.

2. Heat treatment parameter need to mention for fabricated structure.

3. Tolerance of Mitutoyo SJ-301 Surftest instrument is not mentioned while measurement of surface roughness.

4. What size of abrasive used in coarse, medium and fine size particles.

5. What was impact of different speed and concentration on finishing quality of surface?

6. PLOS authors have the option to publish the peer review history of their article (what does this mean?). If published, this will include your full peer review and any attached files.

Reviewer #1: **Yes: **Alberto Boschetto

Reviewer #2: **Yes: **Dr. Jai Kishan Sambharia

---

## [Author Response · Author response to Decision Letter 0]

22 Dec 2022

Dear Editor and reviewers,

We would like to thank you and the reviewers for their efforts in reviewing our work. We carefully went through the reviewers’ comments and created an improved version of the paper, based on their comments. We are resubmitting the revised manuscript, together with this accompanying letter answering in detail to each comment. All the modifications introduced in the revision process are marked up using the “Track Changes” in the text.

Kind regards,

The Authors

Comments from the editors and reviewers:

Reviewer 1

The paper focuses on the polishing of SLMed parts by using CDF apparatus. The work shows many lacks in the description of the methodology and the method itself. In following the list of them.

Comment 1

Introduction: some works on the mass finishing applied to SLM parts are neglected. Interesting papers provide basic understanding of the mechanisms and the attainable roughness.

Authors

We thank the reviewer for this suggestion, we have added the following paragraph on this topic in lines 61 - 76 in the Introduction:

“The working principle of MF is based on the use of abrasive polishing media, such as ceramic or plastic beads, agitated by mechanical means, often by rotation or vibration. The polishing media, in some cases combined with a liquid medium, comes into contact with the surface of the submerged parts and removes material from the surface by abrasion, as well as impart a polishing effect. Important processing parameters for MF processes are the size of the polishing media, speed of motion, and processing time. Generally, the use of larger sized particles results in a higher material removal rate and a better surface quality (14,15). Prolonged contact with the abrasives in the form of a longer processing time results in the same effect (14,16,17), up to a limit (15). Higher speeds are also associated with higher material removal rates, due to higher forces being applied to the workpieces (16,18,19). The fill level of the container with polishing media was shown to have a negligible effect on material removal rates, higher load levels decreased the chance of contact between the abrasives and workpiece (14,18). The advantages of MF processes are that the process is hands-off, it can process multiple parts at once, and it can be customized to the type of metal and surface finish required. 

When the settings of MF processes are properly selected, surfaces with Ra of 0.52 to 5 µm Ra can be obtained (14). …”

Comment 2

Material and Methods: since the previous observation, the line 103 should be revised adding information coming from the references.

Authors

We thank the reviewer for this suggestion. Due to COVID restrictions at the time, we were unable to use our own facilities, and instead we outsourced the production of the parts to an external company. As such, the specific settings used for the SLM printer, such as hatch distance and laser intensity, were not disclosed since they are proprietary information of the commercial company. As an alternative we gave the surface roughness value, which gives an indication of the initial roughness of the part with which we started. The surface roughness corresponds to values commonly encountered in SLM studies. However, we do not feel comfortable speculating towards the precise settings of the SLM process used. We have clarified in the revised manuscript in lines 113 - 114 the reason for not giving the values: 

“The specific print and processing settings used for the test parts were not disclosed by the manufacturer.”

In addition, in lines 177 - 178, we added references to studies with similar surface roughness values: 

“The initial surface roughness on the side of the test part was Ra = 9.49 µm before polishing, which is in line with reported roughness values between 2 and 15 µm Ra for metal printed parts (14,28).”

Comment 3

Material and Methods: the use of lacquer leaves many doubts about the media to surface interactions at the beginning of the operation. The CDF is a delicate operation with relatively low energy and activation mechanisms (chipping, plastic deformations, microcracking, microfatigue) may be plagued, modified or postponed.

Authors

This is indeed true. We acknowledge that the presence of the lacquer influences the results of the CDF process. Therefore, our study was not meant to speculate about the efficiency or suitability of the CDF process, but rather an attempt to gain more insights into the effect of the process on various types of features. The influence of the lacquer on the obtained results depends on many factors, which we feel are outside the scope of this study. Rather, we hope to have shown a practical approach towards gaining useful insights into design guidelines to implement CDF on SLM-ed parts. The above is addressed in lines 147 - 148: 

“No additional surface roughness measurements were performed, since the applied lacquer would interfere with the measured roughness values.”

As well as in the section Limitations and Recommendations, lines 259 - 264: 

“…the presence of the lacquer on the part will have influenced the effectiveness of the polishing process to some extent. Although the lacquer is less hard than the stainless steel of which the parts are made, this is still an additional layer that needs to be polished away. Therefore, it is an effective measure of visualizing the polishing patterns, but less suitable to determine the actual efficiency of the polishing process with regards to the surface roughness and the material removal rate.“ 

In addition, we clarified line 94 in the Introduction, and added lines 125-127 in Materials and Methods: 

“It should be noted that since the lacquer influences the effectiveness of CDF, this process will result in less improvements in surface finish than otherwise can be obtained.”

Comment 4

Material and Methods: authors didn’t provide an experimental plan to find polishing parameters such as: media shape size and material, compound amount and type, rotational speed, water flow, filling percentage, etc. The manufacturer information and experience are generally related to 316L parts fabricated via traditional technologies and different results are expected on the SLMed material completely different in microstructure.

Authors

We agree with the reviewer, however we do feel that an experimental plan to find the optimal polishing parameters is out of the scope of this study. The goal of our study was not to find the optimal polishing parameters for the test part, or to obtain a perfectly polished part, but rather to collect information to indicate the effect of CDF on certain features of complex parts. The recommended manufacturer settings have been applied in preliminary tests, however were not optimized for this specific test part. 

We clarified this in the manuscript by adding the following in lines 95 - 97 and 101 -102 in the Introduction:

“Therefore, in this study, we investigate the influence of centrifugal disk finishing (CDF), a common MF process, on the geometrical features of a complex, 3D printed part, in order to obtain information that can be used in the design process. Since the effectiveness of CDF on surface improvements has already been proven, we did not optimize the settings of CDF for our part. Instead, we investigate how geometrical features are altered as a result of the process, and which types of features cannot be accessed by CDF. In order to do so, we designed an experiment in which a 3D printed test part was coated with a blue marking lacquer for metals, before undergoing a number of different polishing steps. After polishing, the remaining lacquer gives a visual pattern of the surfaces on the part to which the polishing media had access. The results led to a number of design guidelines that can be of help during the design process of SLM parts.”

In Limitations and Recommendations, lines 264 - 268 we added the following:

“Additional research to optimize the processing settings of CDF could improve the obtainable results, for instance by comparing the influence of different rotational speeds and media sizes on the geometrical features. We recommend that in order to optimize the processing settings for a specific design, test parts with representative features of the final design should be used (29).”

Comment 5

Results and discussion: The previous observation can explain the Ra results of this work. The obtained values after the polishing reveal that the selected processing parameters are not really efficient with respect to other mass finishing operations applied to the same SLM material.

Authors

This is indeed true, however we cannot discount the effect of the influence of the lacquer, which had to be polished away first, and led to a worse surface roughness than otherwise could have been obtained. Many other studies, as noted in Comment 1, have shown the potential of CDF with optimum parameters. We are fully aware that the approach shown in this study would not lead to optimal processing settings due to the influence of the lacquer, but it was used for the generation of design guidelines that can be applied to parts that should be polished using CDF. We refer to our response to Comment 3 and 4 as to the alterations we made in the manuscript to clarify this point. 

Comment 6

Results and discussion: The assessment at line 190 should be verified.

Authors

We thank the reviewer for this observation, we have nuanced the statement (line 209) to:

“This is in line with other research suggesting larger particles are responsible for most of the material removal (14,15).”

Comment 7

Material and Methods: The experimental campaign is poor and extraction at intermediate times is general requested for a deeper investigation of the surface evolution. The image processing technique lacks in repeatability with the following issues. The using of a Photoshop algorithm does not allow a clear understanding of the RGB channels selection criteria. Moreover, when a color selection is adopted, particular care must be paid on the light source which may modify the radiation reflection spectrum and have local and surrounding undesired effects on the under-detection feature. Finally, no indication or motivation is provided about the selection of the threshold in Matlab.

Authors

We agree with the reviewer that extraction at intermediate times is generally requested for a deeper investigation of the surface evolution, however this was not the goal of our study. Rather we wanted to illustrate the effects of CDF on geometrical features, since many other studies have attributed themselves to investigating the surface evolution, as pointed out in Comment 1. 

The image processing technique is indeed not perfect. However, we took care to ensure the process is repeatable by giving as many specifications as possible. The Photoshop process used to extract the blue tones is based on an algorithm built into the software, we have not used the option that lets us select the RGB channels manually. Therefore, for photos taken in similar lighting circumstances results of the tool will be comparable and the algorithm will be repeatable. We have clarified this in Line 160 - 161: 

 “The images were processed in Photoshop (Adobe Inc., USA) to extract only the blue tones, using the ‘Color Range’ tool and then selecting ‘Blues’, which automatically selects the blue pixels in the image.” 

For the photos, we took care that the lighting and source were the same for each image, although we can indeed not guarantee that this result is 100% reproducible. Since the lighting conditions were the same for all images, we feel comfortable comparing the results with each other, as potential reflection effects will occur in the same amount for all images. We have added the lighting conditions in which the images were taken in Line 158.

The threshold in Matlab was used because of previous results obtained with similar scripts, for which we have now added the references in Line 163. 

Reviewer 2

In this paper authors used the centrifugal disk finishing process for finishing of o improve the quality of 3D printed surfaces. The authors designed and 3D printed test parts with different features using selective laser melting, which were coated with a blue metal lacquer prior to polishing. The surface improvement is observed through Mitutoyo SJ-301 Surftest instrument.

Comment 1

The authors have not added the image of CDF instrument.

Authors

We thank the reviewer for this suggestion, we have added the image in Figure 2c. 

Comment 2

Heat treatment parameter need to mention for fabricated structure.

Authors

Unfortunately, since we had to rely on a commercial company due to COVID restrictions at the time, we do not have the exact heat treatment parameters, since this is proprietary information of the company. We clarified this in Line 113 - 114:

“The specific print and processing settings used for the test parts were not disclosed by the manufacturer.”

Comment 3

Tolerance of Mitutoyo SJ-301 Surftest instrument is not mentioned while measurement of surface roughness.

Authors

We thank the reviewer for this suggestion, we have added the tolerance in line 116: 

“The surface roughness of the flat side on one of the test parts was measured (Mitutoyo SJ-301 Surftest, Mitutoyo Corporation, Japan) with a tolerance of 0.01 µm …” 

Comment 4

What size of abrasive used in coarse, medium and fine size particles.

Authors

The polishing particles are shown in Figure 2d with a ruler, for clarity we have also added the sizes in Table 1, row 3: 

 Coarse step Medium step Fine step

Polishing media DZS 6/6 KM 10 and PM 10 ZSP 3/5 and DZP 3/3 SK

Size 6 x 6 mm 10 x 10 mm and 10 x 10 mm 3 x 5 mm and 3 x 3 mm

Compound SC15 SC15 SC5

Speed 280 rpm 260 rpm 220 rpm

Water flow 10 L/h 10 L/h 10 L/h

Water concentration 3% 3% 3%

Type Ceramic-bonded Plastic-bonded Ceramic-bonded

Comment 5

What was impact of different speed and concentration on finishing quality of surface?

Authors

In this study, we did not research the optimum processing parameters or their influence on the surface quality. Rather, we focused on the influence of the different steps on the geometrical features. However, previous research has indicated the influence of various process parameters on the surface quality. We have added this information in Lines 61 – 76:

“The working principle of MF is based on the use of abrasive polishing media, such as ceramic or plastic beads, agitated by mechanical means, often by rotation or vibration. The polishing media, in some cases combined with a liquid medium, comes into contact with the surface of the submerged parts and removes material from the surface by abrasion, as well as impart a polishing effect. Important processing parameters for MF processes are the size of the polishing media, speed of motion, and processing time. Generally, the use of larger sized particles results in a higher material removal rate and a better surface quality (14,15). Prolonged contact with the abrasives in the form of a longer processing time results in the same effect (14,16,17), up to a limit (15). Higher speeds are also associated with higher material removal rates, due to higher forces being applied to the workpieces (16,18,19). The fill level of the container with polishing media was shown to have a negligible effect on material removal rates, higher load levels decreased the chance of contact between the abrasives and workpiece (14,18). The advantages of MF processes are that the process is hands-off, it can process multiple parts at once, and it can be customized to the type of metal and surface finish required.”

---

## [Decision Letter · Decision Letter 1]

16 Jan 2023

PONE-D-22-29565R1Polishing of metal 3D printed parts with complex geometry: visualizing the influence on geometrical features using centrifugal disk finishing

PLOS ONE

Dear Dr. Lussenburg,

Thank you for submitting your manuscript to PLOS ONE. After careful consideration, we feel that it has merit but does not fully meet PLOS ONE’s publication criteria as it currently stands. Therefore, we invite you to submit a revised version of the manuscript that addresses the points raised during the review process.

We look forward to receiving your revised manuscript.

Kind regards,

Amitava Mukherjee, ME, Ph.D.

Academic Editor

PLOS ONE

Reviewers' comments:

Reviewer's Responses to Questions

**Comments to the Author**

1. If the authors have adequately addressed your comments raised in a previous round of review and you feel that this manuscript is now acceptable for publication, you may indicate that here to bypass the “Comments to the Author” section, enter your conflict of interest statement in the “Confidential to Editor” section, and submit your "Accept" recommendation.

Reviewer #1: All comments have been addressed

Reviewer #2: All comments have been addressed

2. Is the manuscript technically sound, and do the data support the conclusions?

Reviewer #1: Partly

Reviewer #2: Yes

3. Has the statistical analysis been performed appropriately and rigorously? 

Reviewer #1: No

Reviewer #2: Yes

4. Have the authors made all data underlying the findings in their manuscript fully available?

Reviewer #1: Yes

Reviewer #2: Yes

5. Is the manuscript presented in an intelligible fashion and written in standard English?

Reviewer #1: Yes

Reviewer #2: Yes

6. Review Comments to the Author

Reviewer #1: Comment 1 was well done. In #2 I understand your condition. In #3 I agree with some extends. I don't agree with the answer t ocomment #4: since a manufacturing process is applied, it is important to deepn how the processing paramaters the author selected, both in term of factors and levels, affect the results; the sentence "we investigate the influence of centrifugal disk finishing (CDF), a common MF process, on the geometrical features of a complex, 3D printed part," highligths the use of a traditional technique on a non traditional part (AMed material); I am not requesting an optimization, rather the investigation the autthors mention. This point could help to provide observation on comment #5. Comment #6 ok with some extends. Comment #7: the authors should check what is provided by software they cannot control. In a scientific research the results must be sure as well as the methods to retrieve them.

Reviewer #2: The authors have included all suggestions in revised article. Authors are advised to do more experimentation to optimize the process variable in future research work.

7. PLOS authors have the option to publish the peer review history of their article (what does this mean?). If published, this will include your full peer review and any attached files.

Reviewer #1: **Yes: **Alberto Boschetto

Reviewer #2: **Yes: **Dr. Jai Kishan Sambharia

---

## [Author Response · Author response to Decision Letter 1]

3 May 2023

Dear Editor and reviewers,

Hereby we submit our revised manuscript in response to the reviewers’ comments. We are resubmitting the revised manuscript, together with this accompanying letter answering in detail to each comment. All the modifications introduced in the revision process are marked up using the “Track Changes” in the text. Hopefully the responses below clarify any confusion about this study.

Kind regards,

The Authors

Comments from the editors and reviewers:

Reviewer 1

Comment 4 response:

I don't agree with the answer t ocomment #4: since a manufacturing process is applied, it is important to deepn how the processing paramaters the author selected, both in term of factors and levels, affect the results; the sentence "we investigate the influence of centrifugal disk finishing (CDF), a common MF process, on the geometrical features of a complex, 3D printed part," highligths the use of a traditional technique on a non traditional part (AMed material); I am not requesting an optimization, rather the investigation the autthors mention. This point could help to provide observation on comment #5.

Authors

The settings that were recommended by the manufacturer were in this case for 3D printed parts, not for traditionally manufactured parts. This recommendation was based on the coarse nature of unfinished SLM’ed parts, and was tested on a part before starting this test. There are still many more investigations necessary to determine the effect of all processing parameters on AM-ed parts. Also, the selection of processing parameters depends on the required result, within the requirements of the design. In this case that can explain the less effective result compared to other mass finishing operations, as was pointed out in comment 5. 

Since there is already a lot of research into how the process parameters of CDF influence the surface characteristics for SLM’ed parts, we chose fixed process parameters and focused on a few differences, i.e. duration of polishing time and coarseness of the media, to investigate which effect these have on the geometrical features. We clarified the above and explain that we focus on the coarseness of the media and the total polishing duration (lines 128-132):

“In a preliminary test, a polishing schedule consisting of three steps of 120 minutes with coarse, medium, and fine polishing media was tested on a simple part, as advised by the manufacturer of the CDF for parts produced using SLM. In Table 1, the details of the used process steps are given, for each step fresh media was used. We investigated the influence of the coarseness of the media and the total duration of polishing on the geometrical features, by dividing the twelve parts into four groups of three that were polished in different polishing steps, as shown in Table 2.”

We also addressed this in the recommendations and limitations section (lines 261-264):

“Additional research into other process parameters of CDF, such as rotational speed, is required to provide insight into their effect on geometrical features. We recommend that in order to optimize the processing settings for a specific design, a test part can be used with representative features as used in the final design (29).”

Response Comment 7: the authors should check what is provided by software they cannot control. In a scientific research the results must be sure as well as the methods to retrieve them.

Authors:

We believe there is no part of the software that is out of our control, in fact, the way in which we used the software ensures that the results are the same for all images and can be compared. Adobe Photoshop is a common tool used in image processing, similar approaches are for instance used in references [1-3] below (but many more can be found), which show the Color Range command is reliable in selecting a specific range of colour for the purpose of image processing. We added a brief explanation on the working of the color range command in the Methods section (lines 156-159), although we feel expanding on this method more is out of scope for this study: 

“The images were processed in Photoshop (Adobe Inc., USA) to extract only the blue tones. An RGB image contains three channels: red, green, and blue. The ‘Color Range’ command allows the user to select pixels of an image based on their colour channel. By selecting ‘Blues’ within the Color Range command, only pixels within the blue channel were selected.”

[1] Dahab et al. (2004). Digital quantification of fibrosis in liver biopsy sections: Description of a new method by Photoshop software. Journal of Gastroenterology and Hepatology, 19(1), 78–85 

[2] Chen, Li, and Gao (2018). The Colony Count Based on Image Processing Using Matlab and Photoshop. Advances in Computer Science Research, volume 80

[3] Stewart, A. M., Edmisten, K. L., Wells, R., & Collins, G. D. (2007). Measuring Canopy Coverage with Digital Imaging. Communications in Soil Science and Plant Analysis, 38(7-8), 895–902.

---

## [Decision Letter · Decision Letter 2]

14 Jun 2023

PONE-D-22-29565R2Polishing of metal 3D printed parts with complex geometry: visualizing the influence on geometrical features using centrifugal disk finishingPLOS ONE

Dear Dr. Lussenburg,

Thank you for submitting your manuscript to PLOS ONE. After careful consideration, we feel that it has merit but does not fully meet PLOS ONE’s publication criteria as it currently stands. Therefore, we invite you to submit a revised version of the manuscript that addresses the points raised during the review process.

We look forward to receiving your revised manuscript.

Kind regards,

Amitava Mukherjee, ME, Ph.D.

Academic Editor

PLOS ONE

Journal Requirements:

Reviewers' comments:

Reviewer's Responses to Questions

**Comments to the Author**

1. If the authors have adequately addressed your comments raised in a previous round of review and you feel that this manuscript is now acceptable for publication, you may indicate that here to bypass the “Comments to the Author” section, enter your conflict of interest statement in the “Confidential to Editor” section, and submit your "Accept" recommendation.

Reviewer #1: All comments have been addressed

2. Is the manuscript technically sound, and do the data support the conclusions?

Reviewer #1: Partly

3. Has the statistical analysis been performed appropriately and rigorously? 

Reviewer #1: Yes

4. Have the authors made all data underlying the findings in their manuscript fully available?

Reviewer #1: No

5. Is the manuscript presented in an intelligible fashion and written in standard English?

Reviewer #1: Yes

6. Review Comments to the Author

Reviewer #1: I invite again the authors to motivate the selection of the processing parameters (they told the manufacturer suggested them). In particular some indications must be given for the selection of the geometry which is different for the three levels of charge and obviusly they will affect the contribution to the outcomes. Please detail the color selection (numerically which rules is given to select the blue color) and the sensitivity of the procedure to the lighting by providing a sort of calibration which considers the amount of pixels after im2bw function.

7. PLOS authors have the option to publish the peer review history of their article (what does this mean?). If published, this will include your full peer review and any attached files.

Reviewer #1: **Yes: **Alberto Boschetto

---

## [Author Response · Author response to Decision Letter 2]

24 Jul 2023

Dear Editor and reviewers,

Hereby we submit our revised manuscript in response to the reviewers’ comments. We are resubmitting the revised manuscript, together with this accompanying letter answering in detail to each comment. All the modifications introduced in the revision process are marked up using the “Track Changes” in the text. 

Kind regards,

The Authors

Comments from the editors and reviewers:

Reviewer 1

Comment #1: I invite again the authors to motivate the selection of the processing parameters (they told the manufacturer suggested them). In particular some indications must be given for the selection of the geometry which is different for the three levels of charge and obviusly they will affect the contribution to the outcomes. 

Authors

The selection of the processing parameters was indeed based on the suggestion of the manufacturer, as we explain in the manuscript. Therefore, we do not have any more information to provide on the selection of the geometry of the charge. For reference, the following studies have used CDF on relatively complex AM parts. Soja et al. [1] used a ceramic angle cut tristar in size 10/12 mm with a runtime of 4 hours to reduce the Ra of their SLM parts from 3.5 – 19.4 to 2.9 µm. Kaynak et al. [2] reduced the Ra from 7 µm to Ra of 2.7 µm after 4 hours with ceramic abrasive particles of undisclosed shape and size. Nicolas et al. [3] used a polymer pyramid of 15 mm and ceramic angle cut triangle in 10/15 mm with a runtime of 1 hour for aluminum and 30 hours for titanium to obtain an Ra of 4.43 µm, down from 10.6, and 3.33 µm, down from 16.3, respectively. Fan et al. [4] used triangular ceramic media of 4/10 mm to obtain an Ra of approximately 4.5 µm after 30 minutes. Lesyk et al. [5] used spherical charge with a diameter of 3 mm for 4 hours and decreased surface roughness by 20%. None give a substantial consideration into the choice of polishing media, and we speculate that the choice is made on a trial-and-error basis or from experience, as is the case for us. 

It can be seen from these studies that there is a great variety in the type and shape of abrasive media chosen, and our choice of media is a combination of the types mentioned above. Consideration of the choice of media depends on the size, shape, and weight of the media, however this cannot be seen separately from other process parameters such as packing of the barrel, speed, type of compound, processing time etc. An investigation into the influence of the polishing media on small, complex parts such as ours would warrant an entire study on its own, and would divert the scope of our current study too much. 

We provided the following argumentation in the manuscript in lines 128 – 135:

“Generally speaking, larger abrasive media will be more effective on the external surfaces of the part and have a faster cut rate, while smaller media are able to reach into the interior regions and small features (11,26). The size of the abrasive media is also important to keep the submerged parts separate from each other and prevent them from clashing (13). The shape of the abrasive media should be chosen in such a way that it permits access to all surfaces of the part (13). Based on preliminary tests and advise of the manufacturer of the CDF, we settled on a polishing schedule consisting of three steps of 120 minutes with coarse, medium, and fine polishing media with different shapes.”

In the discussion, we added the following elaboration in line 269-270: 

“Additional research into other process parameters of CDF, such as rotational speed, and abrasive media shape and size, is required to provide insight into their effect on surface finish as well as geometrical features.” 

Comment #2: Please detail the color selection (numerically which rules is given to select the blue color) and the sensitivity of the procedure to the lighting by providing a sort of calibration which considers the amount of pixels after im2bw function.

Authors

We ran an image with an RGB blue pixel scale ranging in color from 0 to 255, divided in 10 increments, through the Matlab script. The Matlab script was used to calculate the amount of pixels after the im2bw function, which amounted to 80%, corresponding to the 0.8 threshold. We ran this image through the same Photoshop process using the Color Range command and then again through the Matlab script, which resulted in 50% black, indicating that this command includes 62.5% of blue pixels considering the 0.8 threshold. For calibration of the lighting, we photographed one fully coated sample and one fully uncoated sample with the microscope. The uncoated sample amounted to a 0% black output. The coated sample amounted to a 94.7% black output. The difference can be attributed to highlights due to reflection of light which show up as white on the image. However, it should be noted that both coated and uncoated highlights show up white, and therefore the difference cannot be detected, and, since highlights are caused by peaks on the surface, they are more likely to be polished and thus uncoated. Therefore, we did not correct for this value. 

We added the following in the limitations and recommendations, line 266-268: 

“The image analysis method employed in this study is useful for comparison within batches, but due to sensitivity to lighting the absolute results may vary in different conditions.”

---

## [Editor Report · Decision Letter 3]

26 Jul 2023

Polishing of metal 3D printed parts with complex geometry: visualizing the influence on geometrical features using centrifugal disk finishing

PONE-D-22-29565R3

Dear Dr. Lussenburg,

We’re pleased to inform you that your manuscript has been judged scientifically suitable for publication and will be formally accepted for publication once it meets all outstanding technical requirements.

Kind regards,

Amitava Mukherjee, ME, Ph.D.

Academic Editor

PLOS ONE
---

## [Editor Report · Acceptance letter]

4 Aug 2023

PONE-D-22-29565R3 

Polishing of metal 3D printed parts with complex geometry: visualizing the influence on geometrical features using centrifugal disk finishing 

Dear Dr. Lussenburg:

I'm pleased to inform you that your manuscript has been deemed suitable for publication in PLOS ONE. Congratulations! Your manuscript is now with our production department. 

Kind regards, 

on behalf of

Professor Dr. Amitava Mukherjee 

Academic Editor

PLOS ONE